# BENCH-O-MATIC: AUTOMATING BENCHMARK CURATION FROM CROWDSOURCED DATA

## ABSTRACT

The rapid evolution of Large Language Models (LLMs) has outpaced the development of model evaluation, highlighting the need for continuous curation of new, challenging benchmarks. However, manual curation of high-quality, human-aligned benchmarks is expensive and time-consuming. To address this, we introduce Bench-O-Matic, an automated pipeline that leverages LLMs to curate high-quality, open-ended prompts from large, crowd-sourced datasets, enabling continuous benchmark updates without human in the loop. We apply Bench-O-Matic to datasets such as Chatbot Arena and WildChat-1M, extracting challenging prompts and utilizing LLM-as-a-Judge for automatic model evaluation. To validate benchmark quality, we propose new metrics to measure a benchmark's alignment with human preferences and ability to separate models. We release Eval-O-Matic, a benchmark consisting 500 challenging prompts curated by Bench-O-Matic. Eval-O-Matic provides 3x higher separation of model performances compared to MT-Bench and achieves 98.6% correlation with human preference rankings, all at a cost of $20. Our work sets a new framework for the scalable curation of automated benchmarks from extensive data.

## 1 INTRODUCTION

The proliferation of Large Language Models (LLMs) has spurred advancements as models expand their capabilities by training on increasingly vast and diverse datasets. Traditional static benchmarks (Wang et al., 2019; Rajpurkar et al., 2016; Bowman et al., 2015; Dolan & Brockett, 2005; Bos & Markert, 2005; Hendrycks et al., 2021a) are quickly becoming saturated and struggle to differentiate state-of-the-art models.

To address these limitations, recent benchmarks like GPQA (Rein et al., 2023) source high-quality and challenging prompts from domain experts. Although these efforts have produced challenging evaluation sets, they come at a steep price—GPQA, for instance, cost over $120,000 to curate its 500 multiple-choice questions (Rein, 2024). The reliance on manual curation makes such benchmarks difficult to produce. Moreover, their static nature is susceptible to test-set leakage and overfitting as models are trained on similar datasets. This necessitates the continuous development of new benchmarks, exacerbating the cost and labor of manual curation. Further, many of these benchmarks rely on close-ended tasks that fail to capture the open-ended nature of real-world interactions, undermining their cost-effectiveness for evaluating alignment to user preference.

An alternative approach without manual curation involves crowdsourcing prompts through live evaluation platforms such as Chatbot Arena (Chiang et al., 2024). These platforms test models against a continuous stream of fresh, open-ended queries and user feedback. However, real-time human evaluation is both expensive and time-consuming, rendering these platforms infeasible for frequent evaluations by model developers. Moreover, while the crowd-sourced prompts represent real-world and open-ended tasks, their quality varies in difficulty and cannot be converted to challenging benchmarks without careful data filtering.

In light of these open challenges, there is a pressing need for an automated pipeline which can curate high-quality prompts dynamically at scale. In this paper, we introduce Bench-O-Matic, an automated benchmark curation system designed to address these gaps. Bench-O-Matic leverages LLMs to curate, filter, and validate prompts based on seven indicators of high-quality prompts, such as specificity and

| | Evaluation | Open-Ended | Prompt Curation | Prompt Source |
|---|---|---|---|---|
| Eval-O-Matic | Automatic | Yes | Automatic | Configurable |
| MMLU, MATH, GPQA | Automatic | No | Manual | Fixed |
| MT-Bench, AlpacaEval | Automatic | Yes | Manual | Fixed |
| Live Bench, Live Code Bench | Automatic | No | Manual | Fixed |
| Chatbot Arena | Human | Yes | Crowd-source | Crowd |

Figure 1: Classification of LLM benchmarks: we categorize benchmarks on how the evaluation can be done, whether the evaluated tasks are ground-truth or open-ended, how are the prompts curated, and whether the developer can control the source for the prompts.

domain knowledge, creating a pipeline that can continuously curate benchmarks alongside model development.

We apply Bench-O-Matic to crowd-sourced datasets, both Chatbot Arena (Chiang et al., 2024) and WildChat-1M (Zhao et al., 2024), demonstrating that it can robustly generate high-quality benchmarks that differentiate models. The resulting benchmark, Eval-O-Matic, employs LLM judges (Zheng et al., 2023a; Li et al., 2023) to estimate human preferences against a baseline model, making the entire process—from prompt curation to evaluation—fully automated. We also address potential biases in LLM-based evaluations and propose solutions to mitigate them. To assess benchmark quality, we introduce new metrics that measure a benchmark's ability to confidently separate models and align with human preferences. When compared to leading benchmarks such as AlpacaEval LC (Dubois et al., 2024) and MT-Bench (Zheng et al., 2023a), Eval-O-Matic achieves stronger model separability, tighter confidence intervals, and achieve 98.6% correlation with Chatbot Arena rankings, making it a fast, reliable predictor of downstream model performance.

To summarize, our works makes the following contributions:

1. We propose a novel data curation pipeline, Bench-O-Matic, to automatically construct high-quality benchmarks from crowdsourced data.

2. We propose metrics to capture desired properties in an LLM benchmark, and validate that Eval-O-Matic achieves higher model separation and alignment to human preference than existing benchmarks.

3. We open-source both Bench-O-Matic pipeline and Eval-O-Matic benchmark.

## 2 RELATED WORKS

**LLM benchmarks.** We briefly review widely used LLM benchmarks. Most existing benchmarks are static and ground-truth-based (e.g., multi-choice question answering). They cover a wide range of domains, including math, science, coding, and reasoning. Common ones include MMLU (Hendrycks et al., 2021a), MATH (Hendrycks et al., 2021b), GSM-8K (Cobbe et al., 2021), HumanEval (Chen et al., 2021), DROP (Dua et al., 2019), BigBench (Srivastava et al., 2023), HellaSwag (Zellers et al., 2019), AGIEval (Zhong et al., 2023), GPQA (Rein et al., 2023), as well as comprehensive collection like HELM (Liang et al., 2022). Many have considered task-based evaluation such as IFEval (Zhou et al., 2023), SWE-Bench (Jimenez et al., 2024), BigCodeBench (Zhuo et al., 2024) or AgentBench (Liu et al., 2023). As LLMs become widely adopted in open-ended scenarios involving interaction with humans (e.g., chatbot), many have considered human evaluation using domain experts or crowd raters such as Amazon Mechanical Turk (Karpinska et al., 2021; Wang et al., 2023) to examine models' response quality. As an alternative to human labeling, previous work has shown that LLM-as-a-judge can be effective human preference proxies (e.g., AlpacaFarm (Dubois et al., 2023), MT-bench (Zheng et al., 2023b), AlpacaEval (Li et al., 2023), WildBench (Lin et al., 2024)).

**Benchmark leakage.** A fundamental limitation of static benchmarks is the potential risk of test set leakage (i.e., contamination). Existing works (Carlini et al., 2021; Sainz et al., 2023; Yang et al.,

2023; Reid et al., 2024) have suggested a growing risk of contamination, which undermines the reliability of benchmarks over time, motivating the need for benchmarks that are more frequently updated.

**Live benchmarks.** DynaBench (Kiela et al., 2021) identifies these challenges and recommends creating living and continuously evolving benchmarks. Recent works LiveBench (White et al., 2024), LiveCodeBench (Jain et al., 2024a), MixedEval (Ni et al., 2024), R2E (Jain et al., 2024b), as well as the community based live evaluation, Chatbot Arena (Chiang et al., 2024). However, none of these focus on developing a pipeline for automatic benchmark curation to enable automatic evaluation on open-ended tasks.

## 3 HOW DO YOU MEASURE BENCHMARKS?

We outline two key properties that the benchmark aiming to approximate human preference should possess to provide meaningful comparisons between models:

1. **Separability:** the benchmark should separate models with high confidence.
2. **Alignment with Human Preference:** the benchmark should agree with human preference.

While previous works have focused on alignment, separability is also a crucial consideration when comparing models of similar quality (e.g., different checkpoints from the same training run). However, achieving high-confidence separability is challenging due to limitations in prompt design and inherent variances in LLM evaluations. Overly simplistic prompts fail to distinguish between models, while the randomness in human and LLM judgments leads to inconsistent predictions. As a result, it is often difficult to confidently determine if a model's apparent performance reflects a genuine difference in capability or merely noisy observations, highlighting a need for methods to verify whether a benchmark can reliably separate similar models.

Statistical measures like Pearson (Pearson, 1895) and Spearman Correlations (Spearman, 1961), commonly used in benchmarks such as AlpacaEval (Li et al., 2023) to measure correlation to human preference ranking, may fail to adequately address model separability and ranking instability. In addition, these measures only provide a coarse signal of ranking correlation without quantifying the magnitude of performance differences between model pairs.

To address these shortcomings, we develop three novel metrics: *Separability with Confidence*, *Agreement with Confidence*, and *Pair Rank Brier Score*.

**Separability with Confidence** quantifies the benchmark's confidence by measuring its consistency in predicting the winner of a model pair across random seeds through bootstrapping. This is done by calculating the percentage of model pairs that have non-overlapping confidence intervals of their benchmark scores. A higher percentage indicates that the benchmark is more confident in distinguishing between the performance of different models, as the confidence intervals of their scores do not overlap.

**Agreement with Confidence Interval** measures how well benchmarks A and B confidently distinguish between two models with the same ordering. Given models $\pi_1, \pi_2$, we assign scores based on:

1. If both benchmarks confidently separate $\pi_1, \pi_2$, a score of 1 is assigned if their preference agree, and -1 if they disagree.
2. If either A or B cannot separate $\pi_1, \pi_2$ with confidence, we assign a score of 0.

The final agreement score is the average across all unique model pairs. A score of 1 implies perfect agreement with full confidence, while a score of -1 indicates complete disagreement.

**Pair Rank Brier Score** further assesses an LLM benchmark's capability to predict the ranking of a pair of competing models by rewarding confidence in correct predictions while penalizing confidence when incorrect. Consider two models $\pi_1 > \pi_2$ with disparate quality. Although two benchmarks A and B predict the same ranking $\pi_1 > \pi_2$, they predict $P(\pi_1 > \pi_2)$ as .60 and .90, respectively (undetectable by Spearman correlation). These benchmarks would result in very different Brier scores, reflecting their ability to quantify the magnitude of performance difference between the models.

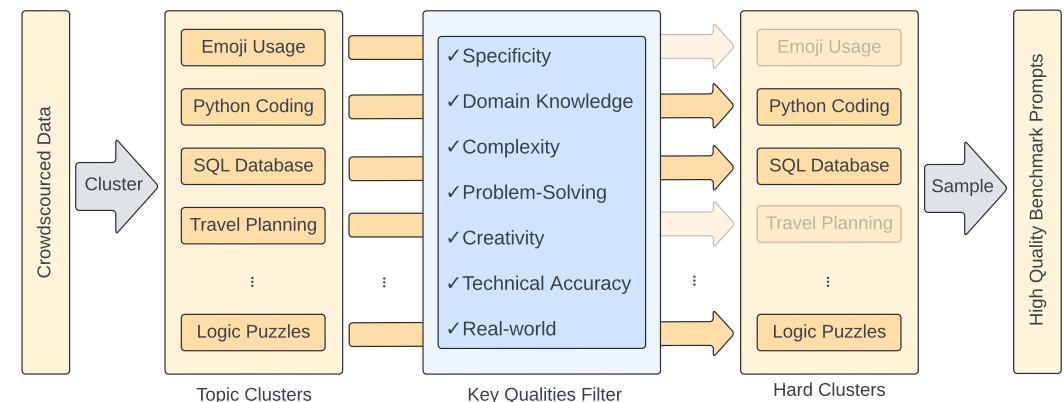

Figure 2: **Bench-O-Matic Pipeline.** Starting with a live data source of crowdsourced user prompts, we first cluster their embeddings to form topic clusters. An LLM annotator then assigns quality scores based on the required skills. Clusters with low quality scores are filtered out, and we sample from the remaining high-quality clusters to create a diverse and challenging dataset of benchmark prompts.

If both benchmarks give the wrong prediction of the winner, we prefer the benchmark with a less confident prediction. In other words, Brier score weighs a benchmark's accuracy and its ability to quantify the appropriate level of uncertainty in its predictions. Background on Pair Rank Brier Score can be found in Appendix A.1.

While no single metric is intended to be individually sufficient, we claim that together, these metrics offer a robust framework for assessing benchmark performance, balancing the need for clear differentiation with alignment to human preferences.

# 4 THE BENCH-O-MATIC PIPELINE AND EVAL-O-MATIC DATASET

## 4.1 BENCH-O-MATIC

The core idea behind how Bench-O-Matic extract high-quality user queries from vast datasets is simple: each prompt is evaluated using a quality score, and prompts with high scores are sampled evenly across diverse topics. Figure 2 illustrates our data creation pipeline.

To identify high-quality prompts, we define seven key qualities that capture the skills necessary to effectively address a query, such as specificity, domain expertise, and creativity (shown in Figure 2). An LLM-based annotator automatically scores each prompt by assessing how many of these qualities are present, producing a "quality score". Detailed instructions for these quality assessments are provided in Section C.

To ensure our filtered prompts span a wide range of tasks, we leverage a topic modeling approach using BERTopic. We first encode each prompt using OpenAI's embedding model, text-embedding-3-small (OpenAI, 2024a), reduce dimensions with UMAP, and apply a hierarchical-based clustering algorithm (HDBSCAN). This process generates distinct topic clusters. Each topic is then summarized and named using an LLM.

Since some topic clusters predominantly contain trivial or poorly defined prompts (e.g., "hi"), we retain only the clusters with high average quality scores and sample prompts evenly across these selected clusters. The resulting dataset consists of mostly well-defined, technical problem-solving queries as required in the above key criteria. Dataset statistics and further details on our filtering and sampling strategy are provided in the following section.

> **Key Prompt Qualities**
>
> - **Specificity:** Does the prompt ask for a specific, well-defined output without leaving any ambiguity?
> - **Domain Knowledge:** Does the prompt test the AI's knowledge and understanding in a specific domain or set of domains?
> - **Complexity:** Does the prompt have multiple components, variables, or levels of depth and nuance?
> - **Problem-Solving:** Does the prompt require active problem-solving: analyzing and clearly defining the problem and systematically devising and implementing a solution?
> - **Creativity:** Does the prompt require a creative approach or solution?
> - **Technical Accuracy:** Does the prompt require an answer with a high degree of technical accuracy, correctness and precision?
> - **Real-world Application:** Does the prompt relate to real-world applications?

## 4.2 EVAL-O-MATIC

We utilize the Bench-O-Matic pipeline to curate 500 challenging benchmark prompts for Eval-O-Matic. Our process begins with an initial pool of 200,000 prompts sourced from Chatbot Arena. We filter out duplicates, multi-turn conversations, and non-English content. Next, we apply hierarchical topic modeling, clustering the prompts into 4,000 distinct topics spanning a diverse range of domains

Then we use GPT-4-Turbo (OpenAI, 2023b) as a judge to assign a "quality score" to each prompt and remove any prompts. Prompts with score less than 6 and topic clusters with mean score less than 5 are discarded, ensuring only the highest quality prompts are retained. The resulting dataset contains over 500 high quality clusters. To construct a 500-prompt benchmark, we sample 2 prompts each from 250 randomly selected clusters. We also ensure the final dataset is free from personally identifiable information or offensive content.

To validate qualities assigned by GPT-4-Turbo, we construct "ground truth" labels for 200 sampled queries by collecting majority votes from GPT-4o (OpenAI, 2024b), Claude-3-Opus, and Gemini-1.5-Pro (Reid et al., 2024). GPT-4-Turbo achieves 85.6% agreement with these labels, demonstrating its reliability as an annotator.

We also applied Bench-O-Matic on 150,000 queries from WildChat-1M (Zhao et al., 2024), which consists of diverse and real-world conversations between users and ChatGPT. Bench-O-Matic identified 185 high quality clusters with 4,500+ prompts. We then randomly sample 2 prompts from each of the highest-quality 125 clusters to create a new benchmark, Wild-O-Matic, which we show to have similar improvement in benchmark quality in section 6.4.

## 4.3 PIPELINE COST AND STATISTIC ANALYSIS

The estimated cost for applying Bench-O-Matic on 200,000 Chatbot Arena queries using GPT-4-Turbo as annotator is approximately \$500 [1]. This cost can be significantly reduced if employing Llama-3-70B-Instruct (Dubey et al., 2024) as annotator instead, which only cost around \$45 [2]. We experimented with Llama-3-70B-Instruct as an alternative annotator and observed similar improvement in downstream benchmark quality. Results are discussed in section 6.4.

Figure 4 illustrates examples of topic clusters across a spectrum of mean scores. Clusters with higher scores correspond to complex topics such as game development or mathematical proofs, while lower-scoring clusters typically involve simpler or ambiguous questions (e.g., "Flirty Texting Strategies"). We provide further examples of prompts and their respective topic clusters in Appendix B.

---

[1] 250 tokens per prompt on average x 200,000 user queries x \$10 per 1 million tokens (OpenAI pricing for GPT-4-1106-Preview).

[2] 250 tokens per prompt on average x 200,000 user queries x \$0.9 per 1 million tokens (TogetherAI pricing, date: 2024-10-01).

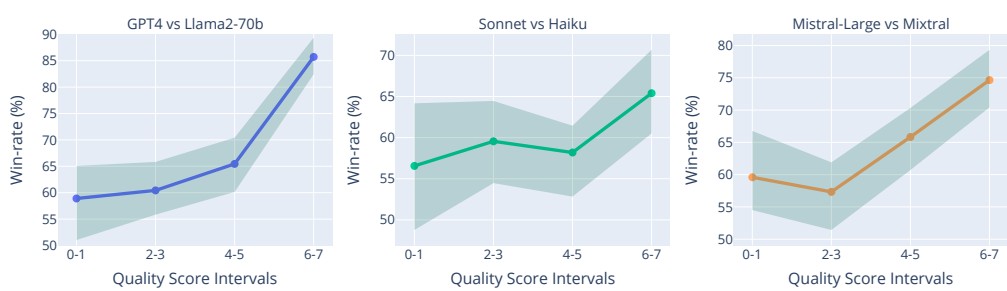

Figure 3: Win-rate of three model pairs (GPT-4-0613 vs Llama-2-70b-chat, Claude-3-Sonnet-20240229 vs Claude-3-Haiku-20240307, and Mistral-Large vs Mixtral-8x7b-Instruct-v0.1) over "quality score". We randomly sample 50 queries for each quality score 0-7 and bootstrap a win-rate and confidence interval between model pairs on each score interval of 2. We observe a similar trend of win-rate between model pairs becomes increasingly separable as the quality score increases.

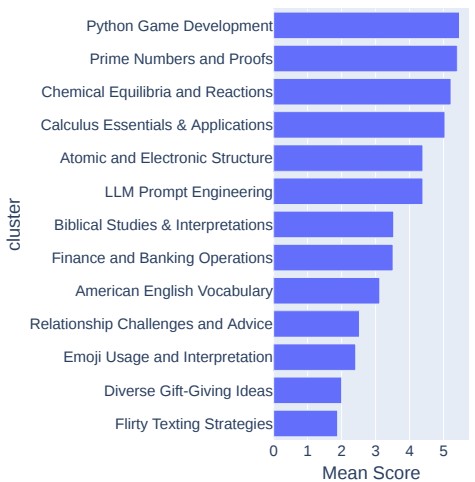

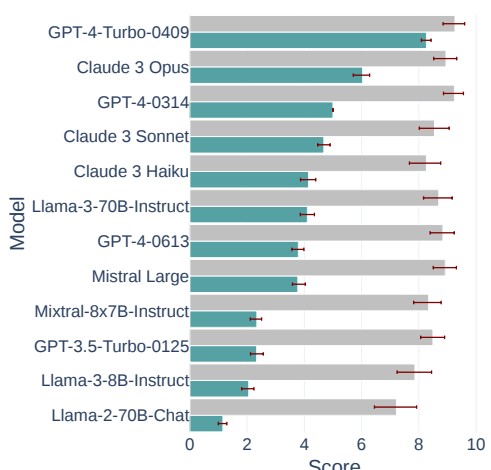

Figure 4: Mean score of various topic clusters in descending order. Higher-scoring clusters correlate to challenging topics. A more complete topic cluster plot is in Figure 6.

Figure 5: Comparison between Eval-O-Matic (Green) and MT-Bench (Grey). The former offers significantly better separability between models and tighter confidence intervals.

To see whether "quality score" assigned during Bench-O-Matic's pipeline correlates with separability and agreement, we sample 50 prompts per score and compare the responses from GPT-4 and Llama-2-70b-Chat (Touvron et al., 2023), with GPT-4-Turbo as judge. In Figure 3 (Left), we observe a strong correlation between high potential score and the win-rate of GPT-4-Turbo over Llama-2-70b-Chat. Similar trends are across other model pairs, including Claude Sonnet vs Haiku and Mistral-Large (team, 2024) vs Mixtral (Jiang et al., 2024a).

## 5 EVALUATION WITH LLM-AS-A-JUDGE

Evaluating models on challenging queries such as Eval-O-Matic requires expert-level judgment due to the depth of domain knowledge and problem-solving skills involved. Expert evaluation, while ideal, is both costly and time-consuming. To address this, we leverage the LLM-as-a-Judge framework (Zheng et al., 2023b; Dubois et al., 2023) as a scalable alternative to approximate human preferences.

We evaluate a model on a given prompt using a pairwise comparison against a strong baseline model (e.g., GPT-4-0314). A judge model (e.g., GPT-4-Turbo or Gemini-1.5-Pro) then scores each output by rating its preference between the pair on a 5-point Likert scale (Likert, 1932) (1 indicates strong

|  | Eval-O-Matic | MT Bench | AlpacaEval 2.0 LC | Chatbot Arena |
|---|---|---|---|---|
| Confidence Agreement | **90.9%** | 26.6% | 82.5% | N/A |
| Separability | **87.4%** | 22.6% | 83.2% | 85.8% |
| Spearman Correlation | **93.2%** | 89.9% | 91.9% | N/A |
| Kendall Tau Correlation | **80.0%** | 64.2% | 77.9% | N/A |
| Brier Score | **0.069** | 0.09 | 0.11 | N/A |
| Real-world | Yes | Mixed | Mixed | Yes |
| Freshness | Frequent Updates | Static | Static | Live |
| Eval cost per model | $20 | $10 | $10 | Very High |
| Prompts per model | 500 | 160 | 800 | 10,000+ |

Table 1: We use a set of top-20 models[3] on Chatbot Arena (2024/04/13) that are also present on the AlpacaEval leaderboard to calculate separability and agreement per benchmark. We consider the human preference ranking by Chatbot Arena (English only) as the reference to calculate agreement.

|  | Wild-O-Matic | Wild-Random-250 |
|---|---|---|
| Confidence Agreement | 88.6% | 36.4% |
| Separability | 86.7% | 75.6% |
| Spearman Correlation | 91.5% | 45.5% |

Table 2: Comparing Wild-O-Matic and a baseline of 250 prompts randomly selected from the WildChat dataset, using GPT-4-Turbo as the judge. Wild-O-Matic has significantly higher separability and agreement to human preference ranking. The experiment demonstrates BenchBuilder's robustness as a general data curation pipeline across different datasets.

preference for model A, 5 indicates strong preference for model B). This scoring method penalizes models more heavily for large losses, effectively distinguishing performance across models. To ensure consistency, we utilize chain-of-thought (Wei et al., 2023) prompting, guiding the LLM judge to generate its own solution before issuing a judgment. Detailed prompt templates are provided in Section C. To avoid potential position bias, we adopt a two-game setup – per query we swap the models on the first and second position. We also study and propose solutions to mitigate potential stylistic biases, such as answer length, and self-bias in LLM-based evaluation in section 6.

This results in 1000 judgments per model evaluation. Following Chatbot Arena, we adopt the Bradley & Terry (1952) model to produce model's the final model scores. We aggregate all pairwise comparisons to the baseline model for all models and bootstrapping the comparisons to retrieve a bootstrapped confidence interval of all models' win-rate against the baseline, producing a ordered ranking of all models by their win-rates.

# 6 EXPERIMENTAL RESULTS

## 6.1 SETUP AND BASELINES

To compare Eval-O-Matic's separability and alignment with humans against other widely used benchmarks, MT-Bench (Zheng et al., 2023b) and AlpacaEval 2.0 Length Controlled (Dubois et al., 2024), we obtain 95% confidence intervals of model performances via applying 100 rounds of bootstrapping on judgment results for each benchmark. For AlpacaEval, we use pre-existing results from their repository. We obtain MT-Bench judgment with no modification to their recommended evaluation setup. For Eval-O-Matic, we employ the system proposed in section 5 by choosing gpt-4-0314 as baseline model for pairwise comparison.

---

[3] gpt-4-turbo-2024-04-09, claude-3-opus-20240229, claude-3-sonnet-20240229, gpt-4-0314 (OpenAI, 2023a), gpt-4-0613, mistral-large-2402, qwen1.5-72b-chat (Team, 2024a), mistral-medium, claude-2.0, gpt-3.5-turbo-0613, claude-2.1, gemini-pro (Gemini et al., 2023), mixtral-8x7b-instruct-v0.1 (Jiang et al., 2024b), gpt-3.5-turbo-0314, yi-34b-chat (AI et al., 2024), tulu-2-dpo-70b (Ivison et al., 2023), dbrx-instruct-preview (Team, 2024b), vicuna-33b (Chiang et al., 2023), starling-lm-7b-alpha (Zhu et al., 2023), llama-2-70b-chat (Touvron et al., 2023)

| | Eval-O-Matic (Style Control) | Eval-O-Matic | AlpacaEval 2.0 LC | MT-Bench |
|---|---|---|---|---|
| Confidence Agreement | **98.6%** | 94.4% | 83.8% | 30.3% |
| Separability | 86.8% | **87.4%** | 83.2% | 22.6% |
| Spearman Correlation | **98.6%** | 94.9% | 88.1% | 90.7% |
| Kendall Tau Correlation | **93.7%** | 85.3% | 70.5% | 77.9% |

Table 3: We apply style control to Chatbot Arena battles (English Hard Prompts) and use its model ranking as reference to calculate alignment. When stylistic confounders like response length are controlled, Eval-O-Matic achieves high alignment to human preferences.

| Model | GPT4-T | Claude3-Opus | Gemini1.5-Pro | Llama3-70B | Ensemble-as-Judges |
|---|---|---|---|---|---|
| Confidence Agreement | 90.9% | 66.7% | 84.8% | 65.6% | **91.5%** |
| Separability | 87.4% | 83.68% | 82.11% | 81.6% | **89.5%** |
| Spearman Correlation | 93.2% | 77.0% | 95.2% | 70.5% | **96.5%** |
| Brier Score | 0.069 | 0.170 | **0.064** | 0.196 | 0.065 |

Table 4: Statistics of Eval-O-Matic with four LLM different judges: GPT4-T (gpt-4-1106-preview), Claude-3-Opus, Gemini1.5-Pro (gemini-1.5-pro-0514), Llama3-70B (llama-3-70b-instruct). We compare rankings produced by these judges against Chatbot Arena (English) ranking (as of 2024/04/13). We observe GPT-4T and Gemini1.5-Pro have higher agreement than Claude-3-Opus and Llama-3-70B. Furthermore, the ensemble of GPT4-T and Gemini1.5-Pro shows even higher agreement.

To ensure fair comparison, we use a set of top-20 models[3] on Chatbot Arena (Chiang et al., 2024) (2024/04/13) that are also presented on AlpacaEval leaderboard (2024/04/13) as ground truth for human preferences on the model ranking orders.

## 6.2 COMPARING SEPARABILITY AND ALIGNMENT ACROSS BENCHMARKS

In Table 1, Eval-O-Matic shows the highest separability (87.4%) against widely adopted LLM benchmarks and offers highest agreement (90.8%) to Chatbot Arena at a $20 cost. In Figure 5, we show Eval-O-Matic offers significantly stronger separability against MT-Bench with tighter confidence intervals. With only 500 prompts, Eval-O-Matic achieve impressive alignment to (and even higher separability than) Chatbot Arena Rankings, which constitutes over 1 million real-world human preferences.

Notably, we observe a significant gap between MT-bench's Spearman Correlation (89.9%) and confidence agreement (22.6%) to Chatbot Arena, an example where Spearman Correlation fails to account for variance of the rankings, and hence cannot adequately measure important ranking granularity of top LLMs. We present a visual comparison between Eval-O-Matic and MT-Bench in Figure 5, highlighting Eval-O-Matic's improved separability.

## 6.3 COMPARING TO A SIMILAR DISTRIBUTION OF HUMAN PREFERENCE

We evaluate Eval-O-Matic with Chatbot Arena's English Hard Prompt leaderboard as ground truth. Since this version of Chatbot Arena leaderboard is based on votes from a more challenging subset of the overall Chatbot Arena battles, we believe it is a more in-distribution comparison for Eval-O-Matic, which also consist of challenging user queries. We observe Eval-O-Matic achieves an overall higher alignment (98.6% Confidence Agreement and 96.7% Spearman Correlation) to human preferences. Results are presented in Appendix Table 9.

## 6.4 ROBUSTNESS AND GENERALIZABILITY

To evaluate the robustness and generalizability of the Bench-O-Matic pipeline, we applied it on 150,000 WildChat (Zhao et al., 2024) dataset and identified 185 high quality clusters with 4,500+ prompts. We then randomly sample 2 prompts from each of the highest-quality 125 clusters to create a new benchmark, Wild-O-Matic. We compare Wild-O-Matic and a baseline of 250 prompts randomly selected from the WildChat dataset in table 2. Results indicates Wild-O-Matic has significantly higher

| No Modification | | Style Control | |
|---|---|---|---|
| Model | Score | Model | Score |
| Llama-3.1-70B-Instruct-detail | 53.5 | Llama-3.1-70B-Instruct | 41.7 |
| Llama-3.1-70B-Instruct-md | 44.9 | Llama-3.1-70B-Instruct-no-md | 39.9 |
| Llama-3.1-70B-Instruct | 44.5 | Llama-3.1-70B-Instruct-detail | 39.8 |
| Llama-3.1-70B-Instruct-chatty | 44.3 | Llama-3.1-70B-Instruct-chatty | 39.5 |
| Llama-3.1-70B-Instruct-no-md | 37.5 | Llama-3.1-70B-Instruct-md | 34.9 |

Table 5: Comparison Between Eval-O-Matic with no modification versus applying style control. Left: Eval-O-Matic with no modification to GPT-4-Turbo judge. Right: style controlled GPT-4-Turbo judge. Asking Llama-3.1-70B-Instruct (Dubey et al., 2024) to response with more detail shows significant performance gain when no style control is applied. However, it is no longer favored with style control. Full table with additional models and system instructions can be found in Appendix Table 6.

separability and agreement to human preference ranking than a random baseline, demonstrating Bench-O-Matic's robustness as a general data curation pipeline for various crowdsourced datasets.

Additionally, we compared Eval-O-Matic against two separate sets of 500 randomly selected prompts from the Chatbot Arena dataset, prior to applying the pipeline extraction. We observe Eval-O-Matic significantly outperforms both random baselines. Results are shown in Appendix Table 7.

To verify whether Bench-O-Matic is not limited to GPT-4-Turbo as annotator for prompt qualities, we employed Llama-3-70B-Instruct as an alternative annotator for prompt curation. We observe the benchmark produced by Llama-3-70b-instruct as the prompt annotator has similar improvement in quality as Eval-O-Matic from random baselines. Results are shown in Appendix Table 8.

### 6.5 MITIGATING STYLISTIC BIASES IN LLM-BASED EVALUATION

LLM-as-a-Judge based evaluation is known to suffer from various biases, such as favoring longer responses (Zheng et al., 2023b; Dubois et al., 2024). AlpacaEval 2.0 Length Control (Dubois et al., 2024) proposes an regression based approach to control length bias in LLM-based evaluation. Chatbot Arena also released a style controlled leaderboard (Li et al., 2024), which attempts to decouple substance from stylistic preferences, including answer length and markdown usage. Following their approaches, we modify how Eval-O-Matic computes the model scores by accounting for the stylistic differences between two answers as additional features to the existing Bradley-Terry model.

We propose controlling for a similar set of stylistic elements used to control human preference on Chatbot Arena for LLM-based evaluation: **answer token length**, density of **markdown headers**, **markdown bold elements**, and **markdown lists**. Technical details on how to extend the Bradley-Terry model for controlling any given style can be found in Appendix A.2.

We apply style control to Chatbot Arena battles and compare the resulting model preference ranking to style controlled Eval-O-Matic, aiming to answer the question: *How well aligned is Eval-O-Matic to human preference when both human preference and LLM judgment are decoupled from stylistic differences?* In Table 3, we show that style controlled Eval-O-Matic achieves **98.6% agreement and correlation** to style controlled human preference ranking, suggesting Eval-O-Matic assessment of model strength separated from style is still highly aligned to humans.

Additionally, we conducted an experiment trying to increase model score on Eval-O-Matic by instructing GPT-3.5-Turbo, Llama-3.1-70b-instruct, and Gemini-1.5-Flash to increase the verbosity and usage of markdown elements in their response and present our results in Table 5. While increasing "detailedness" does increase model performances on Eval-O-Matic when no modifications is applied to GPT-4-Turbo as judge, applying style control is effective at neutralizing this advantage. Our results shows that style controlled model scores cannot be gamed via manipulating response length or markdown usage on Eval-O-Matic. We also observe a reduction in correlation between model score and answer length on Eval-O-Matic. Full results can be found in Appendix Table 12.

## 6.6 MITIGATING SELF-BIASES IN LLM-BASED EVALUATION

LLM-as-a-Judge evaluations are also known to exhibit self-bias. While such biases should manifest as lower alignment with human preferences in our proposed metrics, we conduct a focused analysis to further understand and address this issue. Since Eval-O-Matic uses GPT-4-Turbo as the default judge, we evaluate whether it favors OpenAI models over Anthropic models. Results in Appendix Table 10 indicate that GPT models receive slightly higher average rankings than human preference, while Claude models rank lower.

To reduce this bias, we propose Ensemble-as-Judges, which aggregates judgments from multiple models. The ensemble judges (GPT-4-Turbo and Gemini-1.5-Pro) achieves overall higher separability and alignment with human rankings, as shown in Table 4. Additionally, we also observe that combining GPT-4-Turbo and Gemini-1.5-Pro reduces self-biases. Results can be found in Appendix Table 10. We believe further research into ensemble methods can refine these results and leave this for future exploration.

## 7 LIMITATIONS

While our data sources are drawn from diverse distributions, biases may still exist in our pipeline. For instance, the seven defined qualities may not fully capture the range of possible attributes, potentially skewing towards prompts in technical domains. Furthermore, Eval-O-Matic currently lacks evaluation for multi-turn and non-English interactions due to the limited availability of multi-turn data in crowdsourced datasets and the primary language proficiency of the authors.

To address these limitations, future work will focus on expanding Bench-O-Matic to incorporate multi-turn and multilingual data curation. We also aim to refine our prompt quality definitions, creating a more systematic approach for generating benchmarks that reflect a broader, more inclusive range of scenarios while maintaining high separability and alignment with human judgment. We also plan to explore more advanced version of Ensemble-as-Judges to further enhance our LLM-based evaluation approach.

## 8 CONCLUSIONS

We introduced Bench-O-Matic, a data curation pipeline that transforms crowdsourced data into high-quality benchmarks by seven key qualities. This pipeline enables building challenging and evolving benchmarks which is crucial for evaluating today's advanced language models. Our evaluation metrics, including separability and agreement with confidence, provide a comprehensive assessment of benchmarks. We show the resulting benchmark, Eval-O-Matic, significantly improves separability and alignment with human preferences over existing benchmarks, achieving 98.6% agreement with Chatbot Arena rankings at only $20 per evaluation. We expect Eval-O-Matic to be useful for LLM developers to evaluate their models with confidence and Bench-O-Matic to be a valuable tool for developers seeking to extract high-quality benchmark from vast amounts of data with minimal human effort.

## 9 REPRODUCIBILITY STATEMENT

To ensure reproducibility of our work, we have taken the following steps. We have provided a detailed description of the Bench-O-Matic pipeline in subsection 4.2, with the prompt instruction to the LLM annotator for prompt quality assessment in the Appendix C. The costs associated with running our pipeline and evaluations are provided in subsection 4.3. Our evaluation methodology using LLM-as-a-Judge is explained in section 5, with prompt templates provided in the Appendix C. We have included experiment setups for our ablation studies in section 6. For the appropriate reported metrics and results, we have included confidence intervals obtained through bootstrapping. We will de-anonymize both the Bench-O-Matic pipeline code and the Eval-O-Matic benchmark dataset after decision date. Altogether, researchers should be able to reproduce our results and build upon our work.

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

# A APPENDIX

## A.1 PAIR RANK BRIER SCORE

Bootstrapping is a well-established statistical technique for estimating the distribution of an estimator by sampling with replacement from the original dataset. This approach has become increasingly popular for constructing confidence intervals in LLM leaderboards, such as Chatbot Arena (Chiang et al., 2024). In our proposed evaluation metrics in section 3, such as Separability and Agreement with Confidence Interval, a reliable confidence interval estimation is essential for assessing the performance stability of different models on a given benchmark. Moreover, for metrics like the Pairwise Rank Brier Score, estimating the probability distribution of rank-based model performance is critical. Therefore, applying bootstrapping to the given benchmark provides a straightforward and robust solution for these tasks.

Consider a benchmark consisting of a dataset $D = \{x_1, x_2, \ldots, x_{|D|}\}$ and a scoring function $f$ that measures the performance of $n$ models $\pi_1, \pi_2, \ldots, \pi_n$ on this dataset. Let $D^*$ denote a bootstrap sample of $D$, and let $f(\pi_i, D^*)$ denote the bootstrapped performance score for model $\pi_i$ using the dataset $D^*$. For simplicity, we use $f^*(\pi_i)$ to denote $f(\pi_i, D^*)$.

To use Brier Score (Brier, 1950) for measuring the accuracy of the given benchmark's probabilistic predictions on model performances, we need to compute the forecasted probability that model $\pi_i$ performs lower than $\pi_j$ on the ground truth measurement for every model pair.

$$\hat{P}(f^*(\pi_i) < f^*(\pi_j)) \tag{1}$$

The bootstrapped scores $f^*(\pi_i)$ and $f^*(\pi_j)$ follow an empirical distribution that can be approximated using the Central Limit Theorem (CLT). In most cases, the distribution of $f^*(\pi_i)$ converges asymptotically to a normal distribution, which we also observed in our experiments. Formally, $f^*(\pi_i) \sim \mathcal{N}(\mu_i, \sigma_i^2)$, where $\mu_i$ and $\sigma_i^2$ are the bootstrapped mean and variance, respectively. When this normality assumption does not hold, $\hat{P}(f^*(\pi_i) < f^*(\pi_j))$ can still be estimated from the empirical distribution of the bootstrapped scores.

Let $O_{\pi_i \prec \pi_j}$ denote the ground truth outcome for the model pair $(\pi_i, \pi_j)$, where:

$$O_{\pi_i \prec \pi_j} = \mathbb{1}(\pi_i \text{ performs worse than } \pi_j \text{ on the ground truth evaluation metric}) \tag{2}$$

The Brier Score Loss is then calculated over the benchmark's prediction for each model pair with respect to the ground truth outcome $O$

$$\frac{1}{N} \sum_{\{i,j\}} (\hat{P}(f^*(\pi_i) < f^*(\pi_j)) - O_{\pi_i \prec \pi_j})^2 \tag{3}$$

where $N$ is the number of model pairs.

## A.2 STYLE CONTROL IN MODEL EVALUATION

To mitigate the potential confounding effects of response style on model evaluation, we implemented an enhanced Bradley-Terry regression framework. This method, inspired by recent LLM evaluation technique (Dubois et al., 2024), controls the influence of answer length on judges' preferences. Recently, Chatbot Arena implemented style control (Li et al., 2024) to decouple substance from style in their leaderboard. This approach incorporates style-related features, such as answer length, into the regression model, enabling a distinction between a model's intrinsic capabilities and the influence of these potential confounders like answer style. In essence, style control answers the question: *What would the preference be if everyone has the same style?* This distinction is crucial for a more accurate assessment of model performance without biases.

We extend the standard Bradley-Terry model by introducing additional style features. Let $n$ denote the number of pairwise comparison battles and $M$ the number of models. For each battle $i \in [n]$, we define:

- $X_i \in \mathbb{R}^M$: $X_{i,m} = 1$ if model $m$ is on the presented first to the judge, $X_{i,m} = -1$ if presented last, and 0 otherwise.
- $Y_i \in 0, 1$: The outcome, where 1 indicates the first model won.
- $Z_i \in \mathbb{R}^S$: A vector of $S$ style features for the comparison.

The traditional Bradley-Terry model estimates model strengths $\beta \in \mathbb{R}^M$ through logistic regression:

$$\hat{\beta} = \arg \min_{\beta \in \mathbb{R}^M} \frac{1}{n} \sum_{i=1}^{n} \text{BCELoss}(\text{sigmoid}(X_i^\top \beta), Y_i) \tag{4}$$

Our enhanced model incorporates style coefficients $\gamma \in \mathbb{R}^S$:

$$\hat{\beta}, \hat{\gamma} = \arg \min_{\beta \in \mathbb{R}^M, \gamma \in \mathbb{R}^S} \frac{1}{n} \sum_{i=1}^{n} \text{BCELoss}(\text{sigmoid}(X_i^\top \beta + Z_i^\top \gamma), Y_i) \tag{5}$$

where BCELoss represents the binary cross-entropy loss. We selected the following style features:

- Answer token length
- Density of markdown headers, markdown bold elements, and markdown lists.

For each feature, we compute a normalized difference

$$\text{normalize} \left( \frac{\text{feature}_A - \text{feature}_B}{\text{feature}_A + \text{feature}_B} \right) \tag{6}$$

This normalization technique accounts for the relative difference in features between responses. For instance, the token length difference is normalized as

$$\text{normalize} \left( \frac{\text{length}_A - \text{length}_B}{\text{length}_A + \text{length}_B} \right) \tag{7}$$

We chose this approach over alternatives like the hyperbolic tangent normalization used in AlpacaEval

$$\tanh \left( \frac{\text{length}_A - \text{length}_B}{\sigma(\text{length}_A - \text{length}_B)} \right) \tag{8}$$

Our method better captures proportional differences, especially in cases where absolute differences may be misleading (e.g., 500 vs. 520 tokens compared to 20 vs. 40 tokens).

The resulting $\hat{\beta}$ coefficients represent model strengths controlled for style effects, while $\hat{\gamma}$ quantifies the impact of each style feature on human preferences. To facilitate meaningful comparisons, we normalize the style coefficients. Our analysis revealed that response length was the most influential style factor, with other markdown-related features having secondary effects.

Eval-O-Matic (No Modifications)

| Model | Score | Token # | Header (%) | Bold (%) | List (%) |
|---|---|---|---|---|---|
| gemini-1.5-flash-2-detail | 80.0 | 1035 | 0.010 | 1.503 | 1.288 |
| gemini-1.5-flash-2 | 78.6 | 729 | 0.020 | 1.353 | 1.122 |
| gemini-1.5-flash-2-md | 74.5 | 793 | 0.088 | 1.548 | 1.271 |
| gemini-1.5-flash-2-chatty | 68.2 | 808 | 0.005 | 1.236 | 0.986 |
| gemini-1.5-flash-2-no-md | 61.7 | 574 | 0.003 | 0.924 | 0.979 |
| llama-3.1-70b-detail | 53.5 | 834 | 0.025 | 0.961 | 1.470 |
| llama-3.1-70b-md | 44.9 | 601 | 0.257 | 1.776 | 1.695 |
| llama-3.1-70b | 44.5 | 606 | 0.084 | 0.728 | 1.380 |
| llama-3.1-70b-chatty | 44.3 | 623 | 0.011 | 0.679 | 1.173 |
| llama-3.1-70b-no-md | 37.5 | 522 | 0.010 | 0.123 | 0.986 |
| gpt-3.5-turbo-0125-detail | 25.6 | 416 | 0.008 | 0.447 | 1.540 |
| gpt-3.5-turbo-0125 | 23.1 | 323 | 0.012 | 0.284 | 1.272 |
| gpt-3.5-turbo-0125-md | 22.0 | 328 | 0.372 | 0.877 | 1.601 |
| gpt-3.5-turbo-0125-no-md | 18.0 | 269 | 0.012 | 0.182 | 1.149 |
| gpt-3.5-turbo-0125-chatty | 17.1 | 286 | 0.006 | 0.296 | 1.012 |

Eval-O-Matic (Style Control)

| Model | Score | Token # | Header (%) | Bold (%) | List (%) |
|---|---|---|---|---|---|
| gemini-1.5-flash-2 | 75.5 | 729 | 0.020 | 1.353 | 1.122 |
| gemini-1.5-flash-2-detail | 71.2 | 1035 | 0.010 | 1.503 | 1.288 |
| gemini-1.5-flash-2-md | 69.3 | 793 | 0.088 | 1.548 | 1.271 |
| gemini-1.5-flash-2-no-md | 62.5 | 574 | 0.003 | 0.924 | 0.979 |
| gemini-1.5-flash-2-chatty | 61.5 | 808 | 0.005 | 1.236 | 0.986 |
| llama-3.1-70b | 41.7 | 606 | 0.084 | 0.728 | 1.380 |
| llama-3.1-70b-no-md | 39.9 | 522 | 0.010 | 0.123 | 0.986 |
| llama-3.1-70b-detail | 39.8 | 834 | 0.025 | 0.961 | 1.470 |
| llama-3.1-70b-chatty | 39.5 | 623 | 0.011 | 0.679 | 1.173 |
| llama-3.1-70b-md | 34.9 | 601 | 0.257 | 1.776 | 1.695 |
| gpt-3.5-turbo-0125 | 33.2 | 323 | 0.012 | 0.284 | 1.272 |
| gpt-3.5-turbo-0125-no-md | 30.4 | 269 | 0.012 | 0.182 | 1.149 |
| gpt-3.5-turbo-0125-detail | 28.9 | 416 | 0.008 | 0.447 | 1.540 |
| gpt-3.5-turbo-0125-md | 27.9 | 328 | 0.372 | 0.877 | 1.601 |
| gpt-3.5-turbo-0125-chatty | 27.3 | 286 | 0.006 | 0.296 | 1.012 |

Table 6: Comparison Between Eval-O-Matic with no modification versus applying style control. Prompt for detailed:"You are a helpful assistant who thoroughly explains things with as much detail as possible.", prompt for chatty: "You are a helpful assistant who is chatty.", prompt for md: "You are a helpful assistant who uses as much markdown as possible.", and prompt for no-md: "You are a helpful assistant who never uses markdown." Token represents average number of tokens, header is average markdown header density per token in percentage, bold is average bold markdown element density per token in percentage, and list is average list markdown element per token in percentage.

| Model | Eval-O-Matic | Random Sample 1 | Random Sample 2 |
|---|---|---|---|
| Confidence Agreement | **84.2%** | 57.5% | 66.1% |
| Separability | **80.5%** | 74.7% | 76.3% |
| Spearman Correlation | **94.7%** | 64.7% | 72.5% |
| Brier Score | **0.069** | 0.215 | 0.162 |

Table 7: We compare Eval-O-Matic with two sets of 500 prompts randomly sampled from 75K Chatbot Arena user queries. We evaluate the set of top-20 models and compare various statistics across. Each prompt is judged only once by positioning the baseline answer first.

|                       | Llama-O-Matic | Random 1 | Random 2 | Eval-O-Matic-500 |
|-----------------------|---------------|----------|----------|------------------|
| Confidence Agreement  | 86.0%         | 55.8%    | 58.1%    | 88.4%            |
| Separability          | 84.4%         | 68.9%    | 64.4%    | 88.9%            |
| Spearman Correlation  | 96.4%         | 73.3%    | 70.9%    | 96.4%            |

Table 8: Comparing Llama-O-Matic against two random baselines on 10 of the 20 models outlined in the paper. We observe similar improvement in benchmark quality, suggesting Bench-O-Matic is robust across different choices of LLM annotators.

|                        | Eval-O-Matic |
|------------------------|--------------|
| Confidence Agreement   | 98.6%        |
| Spearman Correlation   | 96.7%        |
| Kendall Tau Correlation| 87.4%        |
| Brier Score            | 0.055        |

Table 9: We compare Eval-O-Matic (gpt-4-1106-preview as judge) to Chatbot Arena Category Hard Prompt (English) on the same set of top-20 models. By comparing Eval-O-Matic to a challenging distribution of queries from Chatbot Arena, we obtain even higher alignment to human preferences.

| OpenAI GPT Series | | | Anthropic Claude Series | | |
|-------------------|-----------|----------|-------------------------|-----------|----------|
|                   | GPT-4-turbo | Ensemble |                       | GPT-4-turbo | Ensemble |
| gpt-4-turbo       | 0         | 0        | claude-3-opus           | 0         | 0        |
| gpt-4-0314        | 1         | 1        | claude-3-sonnet         | -1        | -1       |
| gpt-4-0613        | 0         | -2       | claude-2.0              | -2        | 0        |
| gpt-3.5-turbo-0613| 1         | -1       | claude-2.1              | -1        | 3        |
| gpt-3.5-turbo-0314| 1         | 0        |                         |           |          |
| column average    | 0.6       | -0.4     | column average          | -0.8      | 0.4      |

Table 10: Comparing bias in GPT-4-Turbo as a Judge and Ensemble-as-Judge. We calculate the ranking shift by comparing the human preference ranking (by Chatbot Arena Category Hard Leaderboard) and LLM-judge ranking on OpenAI GPT Series and Anthropic Claude Series. Results show both methods have relatively small shifts, but Ensemble-as-Judge produces a more balanced rank difference than GPT-4-Turbo Judge, suggesting a smaller self-bias than single LLM as a Judge.

| Quality Score | 1+          | 2+              | 3+         | 4+              | 5+         | 6+            | 7+         |
|---------------|-------------|-----------------|------------|-----------------|------------|---------------|------------|
| % of queries  | 95.4        | 83.5            | 61.9       | 48.7            | 33.8       | 17.9          | 0.2        |
| Qualities     | Specificity | Domain-knowledge| Complexity | Problem-solving | Creativity | Tech. Accuracy| Real-world |
| % of queries  | 57.3        | 63.4            | 35.0       | 34.9            | 26.1       | 39.0          | 87.9       |

Table 11: First row is the percentage of queries with quality scores of the column or more in 75K Chatbot Arena data assigned by GPT-3.5-Turbo. Second row is the percentage of queries in 75K Chatbot Arena labeled by GPT-3.5-Turbo with each of the 7 qualities.

| Avg. Token Length | Pearson | Spearman |
|---|---|---|
| No Modification | 0.364 | 0.125 |
| Style Control | 0.193 | -0.025 |

| Naive Verbose Policy | Pearson | Spearman |
|---|---|---|
| No Modification | 0.397 | 0.165 |
| Style Control | 0.231 | 0.028 |

Table 12: Left: Comparing correlation between model score and average token length between GPT-4-Turbo as Judge with no modification versus style controlled. Right: Comparing correlation to model score produced via a "verbose policy", a judge which always picks the longer response. In both cases, style control effectively reduces the correlation to verbosity.

| Model Name | Win Rate | CI Interval | Average Token # |
|---|---|---|---|
| Claude-3-5-Sonnet-20240620 | 79.3 | (-2.1, 2.0) | 567 |
| GPT-4O-2024-05-13 | 79.2 | (-1.9, 1.7) | 696 |
| GPT-4-0125-Preview | 78.0 | (-2.1, 2.4) | 619 |
| GPT-4O-2024-08-06 | 77.9 | (-2.0, 2.1) | 594 |
| Athene-70B | 77.6 | (-2.7, 2.2) | 684 |
| GPT-4O-Mini | 74.9 | (-2.5, 1.9) | 668 |
| Gemini-1.5-Pro-API-Preview | 72.0 | (-2.1, 2.5) | 676 |
| Mistral-Large-2407 | 70.4 | (-1.6, 2.1) | 623 |
| LLaMA-3.1-405B-Instruct-FP8 | 69.3 | (-2.4, 2.2) | 658 |
| GLM-4-0520 | 63.8 | (-2.9, 2.8) | 636 |
| Yi-Large | 63.7 | (-2.6, 2.4) | 626 |
| DeepSeek-Coder-V2 | 62.3 | (-2.1, 1.8) | 578 |
| Claude-3-Opus-20240229 | 60.4 | (-2.5, 2.5) | 541 |
| Gemma-2-27B-IT | 57.5 | (-2.1, 2.4) | 577 |
| LLaMA-3.1-70B-Instruct | 55.7 | (-2.9, 2.7) | 628 |
| GLM-4-0116 | 55.7 | (-2.4, 2.3) | 622 |
| GPT-4-0314 | 50.0 | (0.0, 0.0) | 423 |
| Gemini-1.5-Flash-API-Preview | 49.6 | (-2.2, 2.8) | 642 |
| Qwen2-72B-Instruct | 46.9 | (-2.5, 2.7) | 515 |
| Claude-3-Sonnet-20240229 | 46.8 | (-2.3, 2.7) | 552 |
| LLaMA-3-70B-Instruct | 46.6 | (-2.3, 2.6) | 591 |
| Claude-3-Haiku-20240307 | 41.5 | (-2.5, 2.5) | 505 |
| GPT-4-0613 | 37.9 | (-2.8, 2.4) | 354 |
| Mistral-Large-2402 | 37.7 | (-2.1, 2.6) | 400 |
| Mixtral-8x22B-Instruct-V0.1 | 36.4 | (-2.4, 2.6) | 430 |
| Qwen1.5-72B-Chat | 36.1 | (-2.0, 2.7) | 474 |
| Phi-3-Medium-4K-Instruct | 33.4 | (-2.6, 2.1) | 517 |
| Mistral-Medium | 31.9 | (-1.9, 2.2) | 485 |
| InternLM2.5-20B-Chat | 31.2 | (-2.4, 2.8) | 576 |
| Phi-3-Small-8K-Instruct | 29.8 | (-1.8, 1.9) | 568 |
| Mistral-Next | 27.4 | (-2.4, 2.4) | 297 |
| GPT-3.5-Turbo-0613 | 24.8 | (-1.9, 2.3) | 401 |
| DBRX-Instruct-Preview | 24.6 | (-2.0, 2.6) | 415 |
| InternLM2-20B-Chat | 24.4 | (-2.0, 2.2) | 667 |
| Mixtral-8x7B-Instruct-V0.1 | 23.4 | (-2.0, 1.9) | 457 |
| GPT-3.5-Turbo-0125 | 23.3 | (-2.2, 1.9) | 329 |
| Yi-34B-Chat | 23.1 | (-1.6, 1.8) | 611 |
| Starling-LM-7B-Beta | 23.0 | (-1.8, 1.8) | 530 |
| LLaMA-3.1-8B-Instruct | 21.3 | (-1.9, 2.2) | 861 |
| Snorkel-Mistral-PairRM-DPO | 20.7 | (-1.8, 2.2) | 564 |
| LLaMA-3-8B-Instruct | 20.6 | (-2.0, 1.9) | 585 |
| GPT-3.5-Turbo-1106 | 18.9 | (-1.8, 1.6) | 285 |
| Gemini-1.0-Pro | 17.8 | (-1.2, 2.2) | 322 |
| Command-R | 17.0 | (-1.7, 1.8) | 432 |
| Phi-3-Mini-128K-Instruct | 15.4 | (-1.4, 1.4) | 609 |
| Tulu-2-DPO-70B | 15.0 | (-1.6, 1.3) | 550 |
| Starling-LM-7B-Alpha | 12.8 | (-1.6, 1.4) | 483 |
| Gemma-1.1-7B-IT | 12.1 | (-1.3, 1.3) | 341 |
| LLaMA-2-70B-Chat-HF | 11.6 | (-1.5, 1.2) | 595 |
| Vicuna-33B-V1.3 | 8.6 | (-1.1, 1.1) | 451 |
| Gemma-7B-IT | 7.5 | (-1.2, 1.3) | 378 |
| LLaMA-2-7B-Chat-HF | 4.6 | (-0.8, 0.8) | 561 |
| Gemma-1.1-2B-IT | 3.4 | (-0.6, 0.8) | 316 |
| Gemma-2B-IT | 3.0 | (-0.6, 0.6) | 369 |

Table 13: Eval-O-Matic Leaderboard (baseline: GPT-4-0314) with some additional models (Frick et al., 2024; DeepSeek-AI et al., 2024; IGLM et al., 2024; Yang et al., 2024; Cai et al., 2024; Abdin et al., 2024; Team et al., 2024).

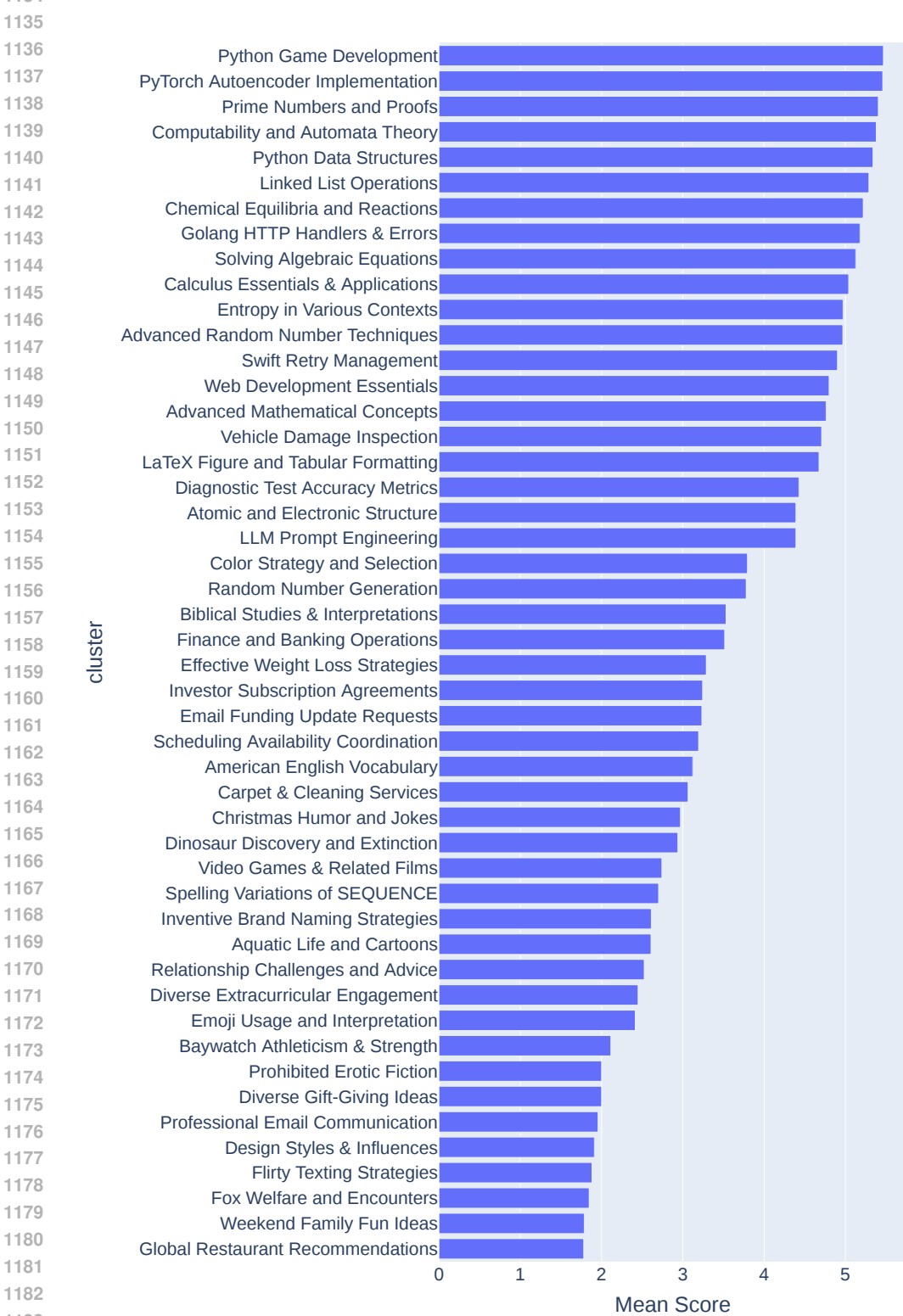

Figure 6: A more complete selection of mean scores of various topic clusters in descending order.

## B EXAMPLES

**Cluster 1: Greetings and Well-Being Inquiry** (Mean Score: 2.7)

```
Yo, what up my brother (Qualities: None)
```

**Cluster 2: US Presidents Query** (Mean Score: 3.2)

```
Who was the president of the US in 1975 (Qualities: Specificity,
Domain-Knowledge, Technical Accuracy, Real-World)
```

**Cluster 3: Physics Problem Solving** (Mean Score: 5.0)

```
A 50,000 kg airplane initially flying at a speed of 60.0 m/s
accelerates at 5.0 m/s2 for 600 meters. What is its velocity after
this acceleration? What is the net force that caused this acceleration?
 (Qualities: Specificity, Domain-Knowledge, Complexity, Problem-
Solving, Technical Accuracy, Real-World)
```

**Cluster 4: OpenCV Image Processing Technique** (Mean Score: 5.5)

```
you are given a task to detect number of faces in each frame of any
video using pytorch and display the number in the final edited video.
(Qualities: All)
```

## C PROMPTS

---

**Prompt Quality Systems Instruction:**

Your task is to evaluate how well the following input prompts can assess the capabilities of advanced AI assistants. For the input prompt, please analyze it based on the following 7 criteria. For each criteria, make sure to explain before determine whether the input satisfy it.

1. Specificity: Does the prompt ask for a specific, well-defined output without leaving any ambiguity? This allows the AI to demonstrate its ability to follow instructions and generate a precise, targeted response.
2. Domain Knowledge: Does the prompt test the AI's knowledge and understanding in a specific domain or set of domains? The prompt must demand the AI to have a strong prior knowledge or mastery of domain-specific concepts, theories, or principles.
3. Complexity: Does the prompt have multiple components, variables, or levels of depth and nuance? This assesses the AI's capability to handle complex, multi-faceted problems beyond simple queries.
4. Problem-Solving: Does the prompt require active problem-solving: analyzing and clearly defining the problem and systematically devising and implementing a solution? Note active problem-solving is not simply reciting facts or following a fixed set of instructions.
5. Creativity: Does the prompt require a creative approach or solution? This tests the AI's ability to generate novel ideas tailored to the specific needs of the request or problem at hand.
6. Technical Accuracy: Does the prompt require an answer with a high degree of technical accuracy, correctness and precision? This assesses the reliability and truthfulness of the AI's outputs.
7. Real-World Application: Does the prompt relate to real-world applications? This tests the AI's ability to provide practical and actionable information that could be implemented in real-life scenarios.

After analyzing the input prompt based on these criteria, you must list the criteria numbers that the prompt satisfies in the format of a Python array. For example, "Criteria Satisfied: [1, 2, 4, 6, 7]".

---

**LLM-Judge System Instruction**

Please act as an impartial judge and evaluate the quality of the responses provided by two AI assistants to the user prompt displayed below. You will be given assistant A's answer and assistant B's answer. Your job is to evaluate which assistant's answer is better.

Begin your evaluation by generating your own answer to the prompt. You must provide your answers before judging any answers.

When evaluating the assistants' answers, compare both assistants' answers with your answer. You must identify and correct any mistakes or inaccurate information.

Then consider if the assistant's answers are helpful, relevant, and concise. Helpful means the answer correctly responds to the prompt or follows the instructions. Note when user prompt has any ambiguity or more than one interpretation, it is more helpful and appropriate to ask for clarifications or more information from the user than providing an answer based on assumptions. Relevant means all parts of the response closely connect or are appropriate to what is being asked. Concise means the response is clear and not verbose or excessive.

Then consider the creativity and novelty of the assistant's answers when needed. Finally, identify any missing important information in the assistants' answers that would be beneficial to include when responding to the user prompt.

After providing your explanation, you must output only one of the following choices as your final verdict with a label:

1. Assistant A is significantly better: [[A>>B]]
2. Assistant A is slightly better: [[A>B]]
3. Tie, relatively the same: [[A=B]]
4. Assistant B is slightly better: [[B>A]]
5. Assistant B is significantly better: [[B>>A]]

Example output: "My final verdict is tie: [[A=B]]".

---

## D ADDITIONAL EXAMPLES

**User:** Use ABC notation to write a melody in the style of a folk tune.

**GPT-4:**

```
X:1 T:Untitled Folk Tune M:4/4 L:1/8
K:G|:G2A2|B2A2|G2E2|D4|E2F2|G2F2|E2C2|B,4|A2B2|c2B2|A2F2|E4|D2E2|F2E2
|D2B,2|C4:|
```

**GPT-3.5-Turbo:** Sure! Here's a simple melody in the style of a folk tune written in ABC notation:

```
X:1 T:Folk Tune M:4/4 L:1/4 K:C
CDE|FGA|BAG|FED| CDE|FGA|BAG|c2z| CDE|FGA|BAG|FED| CDE|FGA|BAG|c2z||
```

This melody is in the key of C major and consists of four measures. Feel free to modify it or add more sections to create your own folk tune!

**Judgement:** My own answer to the prompt would be:

```
X:1 T:My Folk Melody M:3/4 L:1/8
K:D A2|:d2f2a2|b2a2f2|e2d2B2|A4F2| G2A2B2|A2G2F2|E2D2F2|1D6:|2D4A2|
|:d2f2a2|b2a2f2|e2c2A2|B4A2| G2A2B2|A2G2F2|E2D2F2|1D6:|2D4||
```

This melody is in the key of D major, has a 3/4 time signature, and is structured in a typical AAB form common in folk music. It uses a range of an octave, which is typical for folk tunes that need to be easily singable.
Now, let's evaluate the responses from Assistant A and Assistant B.
Assistant A's Answer: Assistant A provides a melody in ABC notation that is coherent and follows the structure of a folk tune. The melody is in 4/4 time, which is common for folk music, and it is in the key of G major. The melody has a two-part form (indicated by the repeat signs |: and :|), which is typical in folk music. The range of the melody is appropriate for a folk tune, and the rhythm is simple and repetitive, which is characteristic of the genre.
Assistant B's Answer: Assistant B also provides a melody in ABC notation that is coherent and could be considered in the style of a folk tune. The melody is in 4/4 time and the key of C major. The melody consists of four measures repeated twice, which is a simple structure that could be found in folk music. However, Assistant B's melody is less traditional in its form compared to Assistant A's, as it does not have a distinct A and B section, which is more common in folk tunes. Additionally, the use of "z" for a half-measure rest is less typical in folk tunes, which usually have a more continuous melodic line.
Comparison: Both assistants provided melodies that could be considered folk tunes. Assistant A's melody has a more traditional structure with a clear A and B section, while Assistant B's melody is simpler and does not follow the traditional AAB or AB form. Assistant A's melody is also more detailed in terms of the ABC notation, providing a full eight-measure section that is repeated, which gives a more complete sense of the tune. Assistant B's melody is correct in its notation but is more basic and less representative of a traditional folk tune structure.
In terms of helpfulness, relevance, and conciseness, both assistants provided relevant and concise answers. However, Assistant A's answer is slightly more helpful due to the more traditional structure of the provided melody.
Creativity and novelty are not particularly applicable in this context, as the task is to create a melody in a traditional style rather than to innovate. Both assistants showed an understanding of the folk tune style, but Assistant A's answer aligns more closely with traditional folk structures.
There are no significant mistakes or inaccurate information in either assistant's answers. However, Assistant B's use of the rest "z" in a folk tune is less conventional.
My final verdict is: Assistant A is slightly better: [[A>B]].

