# OpenReview forum: "Bench-O-Matic: Automating Benchmark Curation from Crowdsourced Data"
_ICLR.cc/2025/Conference — Submitted to ICLR 2025_

### Official Review · Reviewer_5rNy · 2024-10-21

**Soundness:** 3
**Presentation:** 3
**Contribution:** 2
**Rating:** 5
**Confidence:** 3

**Summary:**

The paper proposes approaches to automate the benchmark generation process via the prompting of LLMs. The proposals for different characteristics to establish the baselines are fair and the contributions are around the different scoring mechanisms to 1) evaluate the quality of prompts 2) LLM-based judging of prompt outputs to generate 1-5 score instead of binary preferences and 3) combining them with statistical aggregators to differentiate end evaluate different LLM outputs.

**Strengths:**

The promise of the paper is excellent if delivered -- Reconfigurable automated benchmarks without humans in the loop and via crowd sourced data. With a series of prompting techniques in the pipeline, the approach is fair and well studied. Key innovations are in the design of metrics to separate various models and crux of thesis on generating evaluation data that is of high quality and can be separable.

**Weaknesses:**

The key weaknesses around this paper are the claims that the proposed approach is human-free and easily configurable as shown in the Table comparing the multiple methods. Given that the approach leverages use of ChatBotArena supplied queries  and even though the quality filter will remove the specific poor quality queries, it is not free from the input i.e., humans prompting the different LLMs on the arena and easily being configured to a use case that end users may have in mind. Discussing results on adapting the evaluation framework to beyond what is available in Chat bot Arena would be needed to support the claims of the paper.  Also it would be good to discuss potential biases introduced by using ChatBotArena queries as a starting point. The paper could be strengthened by providing concrete examples or experiments showing how their approach could be adapted to different domains or use cases beyond ChatBot Arena data



An additional area of concern is that almost every step of the pipeline involves a prompt engineering exercise including scoring the final models on a scale of 1-5. This is standard but the question emerges on the fidelity of the LLMs themselves and when they hallucinate themselves. As evidenced by the score sorted by topic cluster, the data does show that for exact answer situations like Python game coding versus loose open ended questions the LLM-judges are not very good.  To strengthen the paper - discuss potential failure modes or biases introduced by relying heavily on LLMs, provide more detailed analysis of how performance varies across different types of questions or topics and suggest ways to mitigate or detect potential hallucinations or errors introduced by LLMs in the pipeline


The details of human annotation were very unclear. See questions below.

**Questions:**

- Is the approach really adaptable/configurable ? Restate the claims if not.
- Can the approach work irrespective of humans in the loop ? i.e., crowd-sourcer providing initial prompts.
- human Annotation study;
    How many human annotators were involved?
    What was the inter-annotator agreement?
    How were discrepancies between annotators resolved?
    Were the human annotators experts in any particular domains?

---

> ### Author Response · Authors · 2024-11-18
> **Official Response by Authors**
>
> We thank the reviewer for their feedback. Below, we address their comments.
>
> > Given that the approach leverages use of ChatBotArena supplied queries and even though the quality filter will remove the specific poor quality queries, it is not free from the input i.e., humans prompting the different LLMs on the arena and easily being configured to a use case that end users may have in mind. Discussing results on adapting the evaluation framework to beyond what is available in Chat bot Arena would be needed to support the claims of the paper. Also it would be good to discuss potential biases introduced by using ChatBotArena queries as a starting point. The paper could be strengthened by providing concrete examples or experiments showing how their approach could be adapted to different domains or use cases beyond ChatBot Arena data
>
> We apologize if we were unclear in our paper: Bench-O-Matic is a data curation pipeline which can identify and select high quality prompts from any large pool of data, not just Chatbot Arena. In Section 6.4, we applied our pipeline to Wild-Chat-1M, which is a dataset consisting of diverse, real-world conversations and is not sourced from Chatbot Arena. We demonstrated that the benchmark curated from Wild-Chat-1M by our pipeline also improved significantly in quality (Line 343). This strengthens the generalizability of our pipeline.
>
> > An additional area of concern is that almost every step of the pipeline involves a prompt engineering exercise including scoring the final models on a scale of 1-5. This is standard but the question emerges on the fidelity of the LLMs themselves and when they hallucinate themselves. As evidenced by the score sorted by topic cluster, the data does show that for exact answer situations like Python game coding versus loose open ended questions the LLM-judges are not very good. To strengthen the paper - discuss potential failure modes or biases introduced by relying heavily on LLMs, provide more detailed analysis of how performance varies across different types of questions or topics and suggest ways to mitigate or detect potential hallucinations or errors introduced by LLMs in the pipeline
>
> We afraid there may be potential misunderstanding. First, we would like to clarify the interpretation of the “score sorted by topic cluster” in Figure 4 as brought up by the reviewer (Line 303). The score for each topic cluster represents the average number of key qualities satisfied by the prompts within the topic cluster. A higher cluster score implies a higher number of key qualities are being satisfied by the prompts within the topic cluster. The score is unrelated to the qualities of the LLM judges. We apologize for the confusion and will make sure to add more descriptions to the figure and improve clarity in the final version of our paper.
>
> As reviewer mentioned, our pipeline employs prompt engineering LLM within our pipeline, prompt selection and model evaluation. In our paper, we validated the fidelity of these components, see below:
>
> **Prompt Selection**: Our data curation pipeline assigns defined qualities to each user prompt in the dataset. For each prompt, we employ GPT-4-Turbo as a judge to determine whether it satisfies each defined quality, generating a quality score based on the number of qualities met. To ensure reliability, we validated GPT-4-Turbo's performance as a judge in our experiment, which demonstrated an 85.6% labeling accuracy (Line 246).
>
> **Model Evaluation**: We employ LLM judges for evaluating models, using the produced model scores to construct a leaderboard. We proposed metrics to measure agreement and correlation to human preference ranking in section 3. Using these metrics, we validated our approach using LLM judgment for model evaluation by demonstrating our benchmark having a strong correlation and agreement with human preference ranking (Line 334).

---

> ### Author Response · Authors · 2024-11-18
> **Official Response by Authors**
>
> ### Reviewer's Questions
>
> > Is the approach really adaptable/configurable ? Restate the claims if not.
>
> As shown in Section 6, our approach is indeed configurable. Bench-O-Matic has three configurable components: an initial dataset, an LLM annotator for prompt selection, and an LLM judge for model evaluation. For the initial dataset, we verified Bench-O-Matic’s effectiveness on two independent datasets, Chatbot Arena and Wild-Chat-1M (Section 6.4). For prompt selection, we experimented with two different LLM judges for annotation, GPT-4-Turbo and Llama-3-70B-Instruct, and demonstrated both produced benchmarks with significantly improved quality (Line 455). For model evaluation, we experimented with four different LLM judges, GPT-4-Turbo, Claude-3-Opus, Gemini-1.5-Pro, and Llama-3-70B-Instruct, and reported their respective agreement and correlation to human preference in Table 4 (Line 393).
>
> > Can the approach work irrespective of humans in the loop ? i.e., crowd-sourcer providing initial prompts.
>
> Our approach automates the process of curating high quality prompts from crowdsourced datasets, which otherwise would require expensive manual prompt selection. Notably, our approach is not limited to Chatbot Arena, but generalizable to any crowdsourced datasets. As detailed in Section 6.4, we demonstrated our generalizability by applying our pipeline on Wild-Chat-1M, which is an independently crowdsourced dataset.
>
> > ​​human Annotation study; How many human annotators were involved? What was the inter-annotator agreement? How were discrepancies between annotators resolved? Were the human annotators experts in any particular domains?
>
> We are concerned there may be potential misunderstandings. In Section 3, we proposed metrics to measure how well a benchmark aligns with human preferences. Our metrics demonstrate that the curated benchmarks strongly agree and correlate with human preference leaderboards (as detailed in Table 1, Line 334). This implies that when evaluating a model, our benchmark can accurately predict its performance on real-world preference tasks, which is the primary goal of an effective benchmark. These leaderboards are based on millions of real-world user preferences across a wide range of tasks and domains. Furthermore, according to the Chatbot Arena paper's user preference analysis, there is a high agreement rate (83%) between user votes and expert assessments. For detailed information about the leaderboard validity and voter analysis, please refer to the Chatbot Arena paper [1].
>
> **Reference**
>
> [1] Chiang et al. “Chatbot arena: An open platform for evaluating llms by human preference”, ICML 2024
>
> ---
>
> We thank the reviewer for their invaluable feedback, and we sincerely hope that the reviewer would reconsider their rating in light of all our points listed in the rebuttal.

---

> > ### Comment · Reviewer_5rNy · 2024-11-18
> >
> > Thanks for the feedback. Given the comments from the authors I think the claims and contributions seem to me misaligned. There seems to be several misunderstandings probably because of the contribution claims.  I would stand by my statement that the approach is not configurable to the use that an end user might have in mind. Secondly, the premise of the work  is that it is dependent on the already collected crowd-sourced data.
> >
> > Minor but over-stating claims: Crowd-source data is human collected (and hence not human free).
> >
> > To ensure that the approach is useful,  an evaluation in a new domain would truly test the system not an independent similar crowd-sourced benchmark.   As such, I will stay with  my review and ratings but curious to see how the other reviewers might respond.
> >
> > The human annotation study points to prior work but I strongly think that the work will benefit from a small study to see if the final benchmark created is agreeable to the application (and if truly configurable) to a broader use case.

---

> ### Comment · Reviewer_3LVj · 2024-11-18
>
> I respectfully disagree with Reviewer 5rNy's interpretation. In my understanding, the paper clearly presents Bench-O-Matic as a *data curation pipeline*. It does not claim that the initial data collection process is human-free, but rather that the curation process itself is automated—a distinction that is emphasized throughout the paper. For example, the word 'curation' is used multiple times in both the abstract (5 occurrences) and the introduction (12 occurrences), highlighting the paper's primary focus. If there are specific parts of the text that gave the impression the data collection itself is human-free, I would appreciate it if you could point them out for further discussion. Personally, I did not interpret it that way.
>
> Beyond this clarification, I find that the claims made in the paper are appropriately framed and well-supported by the evidence provided, even before the authors' rebuttal. Many of the points raised in the questions and weaknesses could be addressed directly by referring to the content already present in the paper. While I initially refrained from addressing these points to allow the authors to respond, I felt the presentation was sufficiently clear from the outset, and the rebuttal only strengthened this impression.
>
> I agree with Reviewer 5rNy that the method relies heavily on prompt engineering, but this is both a standard practice and an unavoidable aspect of this type of research. Importantly, the benchmarks created through this process show strong agreement with human preferences, which, in my view, mitigates this concern.
>
> I believe that the misunderstandings may come from a misreading or an incomplete interpretation of the paper’s focus, rather than from overstated claims by the authors. The distinction between data collection and curation, as well as the evidence supporting the pipeline's configurability, is well-documented in the text. I hope this perspective helps clarify the authors’ intent and highlights the contributions of this work.
>
> Regarding the comment on the human annotation study and the need for a broader evaluation, could you clarify how this ties into the scope of the paper? From my perspective, this work is explicitly focused on curating datasets rather than collecting data. Additionally, I find the evidence for the pipeline’s configurability—such as its successful application to both Chatbot Arena and Wild-Chat-1M datasets—compelling. Using different LLMs for evaluating the prompt and scoring the models also seems relatively simple. Are there specific aspects that led you to question its configurability? I believe this clarification could help align our interpretations.

---

> ### Comment · Reviewer_5rNy · 2024-11-20
>
> Thanks for weighing in.
>
> Is Data curation just chosing a few prompts from the wildchat dataset ? I am trying to triangulate the impact.
>
> In terms of generalization and configurability - how would we adopt this approach to "curate" a benchmark for a 1) scientific agent or 2) answering questions about molecules/proteins and 3) for code-generation tasks ? These are just examples.
>
> I am fairly gracious with my ratings and will stick to them irrespective given the contributions:  As pointed in one of the review above ".... It is a well-executed and solid paper, though not necessarily groundbreaking"; I concur.

---

> ### Comment · Reviewer_3LVj · 2024-11-21
>
> I do agree that the impact is limited in terms of technical research contributions. However, in my opinion, a paper in the 'Datasets and Benchmarks' category should not be expected to make a large technical contribution, but it should provide a tool or methodology that has the potential to support and advance future research. The strength of this work lies in its solid execution, relevance to current research needs, and potential to become a widely used tool.
>
> From my understanding, the generalizability of the pipeline depends on the diversity of the topics in the underlying dataset. If the curated dataset includes questions related to other domains like scientific agents, molecules/proteins, or code generation, the pipeline should adapt and create the appropriate topic clusters, provided the necessary prompts exist. Maybe the authors could expand on this?

---

> ### Author Response · Authors · 2024-11-22
> **Official Response from Authors [1/2]**
>
> We thank the reviewers for their additional comments. We would like to address specific comments from the reviewers first, then summarize key contributions of our work.
>
> > If the curated dataset includes questions related to other domains like scientific agents, molecules/proteins, or code generation, the pipeline should adapt and create the appropriate topic clusters, provided the necessary prompts exist. Maybe the authors could expand on this?
>
> Reviewer 3LVj is correct in noting that our pipeline effectively adapts to create appropriate topic clusters in the source dataset. For example, our pipeline identifies over 4,000 distinct topics from the original dataset from Chatbot Arena across a wide range of domains, such as “Professional Email Communication,” “PyTorch Autoencoder Implementation,” and “Baking and Peanut Butter Recipes.” Then the pipeline evaluates these topic clusters and prompts, selecting only the highest quality clusters and prompts based on the desired specifications. As a result, the final curated benchmark reflects topics that naturally occur in real-world user scenarios while aligning with the specific requirements set by the pipeline. For example, the final curated benchmark, Eval-O-Matic, contains prompts from topic clusters such as “Advanced Algebra and Number Theory”, “PyTorch Autoencoder Implementation”, and “Chess Strategy and Gameplay”. Figure 6 shows a plot with topic clusters and their quality scores assigned by Bench-O-Matic (Page 22).
>
> >In terms of generalization and configurability - how would we adopt this approach to "curate" a benchmark for a 1) scientific agent or 2) answering questions about molecules/proteins and 3) for code-generation tasks ? These are just examples.
>
> We understand Reviewer 5rNy's comment regarding Bench-O-Matic does not produce a benchmark tailored to specific tasks. While we agree that extending the pipeline to curate targeted benchmarks is a promising direction for future work, we believe our approach effectively addresses the current concerns of model developers. Most developers are interested in how well their trained models perform on real-world user queries and the overall user experience post-deployment. This necessitates evaluating model performance on queries sourced from actual user interactions and naturally occurring topic clusters. For example, tasks related to "code-generation" are indeed frequently asked by real users in our crowdsourced datasets, and the final curated benchmarks reflect this and contain topic clusters such as "PyTorch Autoencoder Implementation" and "Web Development & APIs". By focusing on curating benchmarks from crowdsourced data, our pipeline ensures that the resulting benchmarks best reflect the types of questions real users are likely to ask, thus providing a more accurate assessment of a model's real-world performance.
>
> ## Key Contributions
>
> **Bench-O-Matic**
>
> Our pipeline offers a solution to key challenges in the current landscape of LLM evaluation. Traditional static benchmarks often struggle to effectively differentiate state-of-the-art models, fail to align closely with real human preferences, and suffer from issues such as performance saturation and susceptibility to test-set leakage. However, creating new, high-quality benchmarks typically requires extensive manual curation, incurring significant labor costs. Consequently, there is a growing demand among model developers for benchmarks that can:
> 1. Effectively differentiate state-of-the-art models
> 2. Align to real human preferences and evaluate models on real-world user tasks
> 3. A cost-effective pipeline which can constantly curate new, high quality benchmarks to avoid saturation and test-set leakage
>
> To address the first two points, we want to first highlight our novel evaluation metrics: Separability with Confidence Intervals, Agreement with Human Preference, and Brier Score. These metrics provide robust tools to quantify a benchmark’s effectiveness on qualities that matter most to model developers. We believe these metrics empower developers to make informed decisions about which benchmarks best suit their needs. Subsequently, we showed the benchmarks we curated to be effective at differentiating model performances and well-align to human preferences.
>
> To address the third point, our pipeline demonstrated remarkable cost efficiency, processing 200,000 prompts for just \\$45 (Line 259). In contrast, GPQA, which comprised 500 multiple-choice questions, incurred a cost exceeding \\$120,000 (Line 37). This affordability enables Bench-O-Matic to be applied continuously to new datasets, allowing for the curation of high-quality benchmarks on demand. By doing so, it mitigates the risks of saturation and test-set leakage, providing model developers with access to fresh, dynamically curated benchmarks for testing their models.

---

> ### Author Response · Authors · 2024-11-22
> **Official Response from the Authors [2/2]**
>
> **Eval-O-Matic**
>
> Lastly, we expect the benchmark for release to be very useful for the community. From our results provided in Section 6, we demonstrate that the curated benchmarks strongly agree and correlate with human preference leaderboards (as detailed in Table 1, Line 334). This implies that when evaluating a model, our benchmark can accurately predict its performance on real-world preference tasks, which is the primary goal of an effective benchmark, all at a cost of $20.
>
> ---
>
> In summary, we appreciate the reviewers' constructive feedback and take this opportunity to reinforce our contribution to addressing the key challenges in the current landscape of LLM evaluation.
>
> We sincerely hope Reviewer 5rNy would reconsider their rating in light of all our points listed in the rebuttal.

---

> > ### Comment · Reviewer_5rNy · 2024-11-26
> >
> > Thanks. I read all the reviews and rebuttal and will keep my original ratings with a change in the soundness score. I plan to take time off to enjoy the Thanksgiving break in the US and wish you the same if you are celebrating. Good luck.

---

### Official Review · Reviewer_3LVj · 2024-11-04

**Soundness:** 3
**Presentation:** 3
**Contribution:** 4
**Rating:** 8
**Confidence:** 3

**Summary:**

The paper introduces an automated pipeline, Bench-O-Matic, designed to curate prompts and create benchmarks for evaluating large language models (LLMs). The authors propose new metrics to assess benchmark quality, ensuring a clear separability of confidence scores and alignment with human preferences. The prompts are organized into topic clusters to ensure diversity, and an "LLM-as-a-Judge" approach is used to evaluate responses from various LLMs, fully automating the evaluation process. Additionally, the paper presents two novel benchmarks generated using this pipeline: Eval-O-Matic, based on Chatbot Arena, and Wild-O-Matic, derived from WildChat-1M.

**Strengths:**

- The problem statement is clearly defined.
- The paper addresses a significant challenge highly relevant to the current state of AI and places well in the current literature.
- The pipeline is flexible and open-ended, allowing for continuous improvements over time.
- The experiments are comprehensive, demonstrating that the pipeline effectively creates benchmarks based on the metrics defined in the paper, with multiple LLMs evaluated on Eval-O-Matic.
- The paper presents new ideas to evaluate benchmarks to overcome previous issues.

**Weaknesses:**

- Using an LLM to evaluate other LLMs’ responses may limit the complexity of the benchmark prompts. While employing an ensemble of judges partially mitigates this issue, there is still an inherent limitation. However, the advantages of an automated pipeline outweigh this concern, and the authors have implemented techniques to reduce evaluation biases.

I have a hard time finding weaknesses for the paper. It is a well-executed and solid paper, though not necessarily groundbreaking.

**Minor Comments**
- On line 80, "achieve 98.6% correlation" should be "achieve**s** 98.6% correlation".
- On line 82, "Our work**s** makes" should be "Our work makes".
- On lines 206 and 352, "Section C" should probably be changed for "Appendix C" for clarity.
- On line 464, "an regression based approach" should be corrected to "**a** regression-based approach."

**Questions:**

- Previous studies have shown that fine-tuning the LLM-as-a-Judge can significantly improve evaluation robustness. Has this been considered in the current work? This could help improve the quality of the judges, the main limitation of this benchmark.
- In Section 4.2, it states, "We also ensure the final dataset is free from personally identifiable information or offensive content." Could the authors elaborate on how this is achieved? Was this done manually or automatically with the help of an LLM?

---

> ### Author Response · Authors · 2024-11-18
> **Official Response from Authors**
>
> We thank the reviewer for their encouraging feedbacks. We appreciate that the reviewer found our paper sound, well-presented, and relevant to current AI challenges. Below, we address the reviewer's comments and questions.
>
> **Reviewer’s Comments**
>
> We appreciate the reviewer for pointing out all the minor writing corrections. We will make sure to correct them in the final version of our paper.
>
> **Reviewer’s Questions**
> > Previous studies have shown that fine-tuning the LLM-as-a-Judge can significantly improve evaluation robustness. Has this been considered in the current work? This could help improve the quality of the judges, the main limitation of this benchmark.
>
> We appreciate the reviewer's suggestion regarding the fine-tuning of LLM-as-a-Judge. While we haven't yet tested a fine-tuned version on our benchmark, we plan to conduct a comparative study that will evaluate the performance of both fine-tuned and non-fine-tuned LLM judges in the future.
>
> > In Section 4.2, it states, "We also ensure the final dataset is free from personally identifiable information or offensive content." Could the authors elaborate on how this is achieved? Was this done manually or automatically with the help of an LLM?
>
> We verified the final dataset is free from Personally Identifiable information by using Azure PII detection tool. We also used OpenAI moderation API to flagged and remove any prompts with offensive contents. We thank the reviewer for pointing this out and will make sure to detail these steps in the final version of our paper.
>
> ---
>
> We thank the reviewer for their invaluable feedback and hope we have addressed their questions.

---

### Official Review · Reviewer_EWC5 · 2024-11-05

**Soundness:** 2
**Presentation:** 3
**Contribution:** 2
**Rating:** 5
**Confidence:** 2

**Summary:**

The work proposes Bench-O-Matic, a system for automatically curating high-quality, open-ended LLM benchmarks by using large-scale, crowd-sourced data. This tool addresses the need for evolving benchmarks that adapt to the rapid development of LLMs without requiring human intervention.

**Strengths:**

- Bench-O-Matic efficiently creates high-quality benchmarks from crowd-sourced data without human input, the work addressed the scalability issue in benchmark curation.
- The work Introduces novel metrics like Separability with Confidence and Pair Rank Brier Score, enhancing the robustness and reliability of benchmark assessments.
- Eval-O-Matic achieves strong performance alignment with human preferences for only $20 per evaluation, and provides a cost-effective alternative to static benchmarks.

**Weaknesses:**

- Quality insurance. The seven quality criteria may not fully encompass the diversity of user tasks, potentially favoring specific types of prompts over others.
- The synthesis of data is reliant on on LLMs as Judges. The LLM-as-a-Judge framework may introduce stylistic or self-bias, even with adjustments, which could influence benchmark objectivity in certain cases.

**Questions:**

Is there any estimation on the error/quality of the data generated? Or using some metrics to evaluate the similarity of the generated data with the real-world data?

---

> ### Author Response · Authors · 2024-11-18
> **Official Response from Authors**
>
> We thank the reviewer for their feedback. Below, we address their comments.
>
> > The seven quality criteria may not fully encompass the diversity of user tasks, potentially favoring specific types of prompts over others.
>
> As mentioned on line 504, we agree with the reviewer that the seven defined qualities may not fully capture the range of all possible tasks. While these qualities primarily focus on identifying more challenging prompts—particularly those involving problem-solving—we emphasize that Eval-O-Matic demonstrates strong correlation with human preference leaderboards, which are based on millions of real-world user preferences (as shown in Table 5, Line 334). Our pipeline, Bench-O-Matic, is designed to curate prompts from vast datasets based on defined qualities, and is not limited to the seven we identified. Developers can customize their own criteria for desired prompt qualities, such as whether a prompt effectively assesses reasoning abilities.
>
> > The synthesis of data is reliant on on LLMs as Judges. The LLM-as-a-Judge framework may introduce stylistic or self-bias, even with adjustments, which could influence benchmark objectivity in certain cases.
>
> We agree with the reviewer that LLM-as-a-Judge framework may introduce biases (Line 462). In section 4.2, we validated our LLM judges for prompt selection (see “Data Filtering” below). In section 6.5 and 6.6, we addressed and proposed solutions to mitigate stylistic biases and self-biases when using LLM-as-a-Judge for model evaluation (see “Stylistic Bias” and “Self-Bias” section below).
>
> **Data Filtering**
>
> Our data curation pipeline assigns defined qualities to each user prompt in the dataset. For each prompt, we employed GPT-4-Turbo as a judge to determine whether it satisfies each defined quality, generating a quality score based on the number of qualities met. To ensure reliability, we validated GPT-4-Turbo's performance as a judge in our experiment, which demonstrated an 85.6% labeling accuracy (Line 246).
>
> **Stylistic Bias**
>
> We agree with the reviewer that LLM-as-a-Judge may introduce stylistic biases (Line 480). To address this, Section 6.5 presents a standard statistical technique to decouple style and substance in the leaderboard. By isolating stylistic influence from final model scores, the style-controlled benchmark reflects model strength agnostic of style. Notably, this approach removes stylistic confounders rather than arbitrarily adjusting for them. Detailed methodology is provided in the appendix (Page 17).
>
> Furthermore, as detailed in line 441 of Section 6.5, we conducted experiments demonstrating that our style-controlled benchmark cannot be manipulated through stylistic factors—addressing the primary concern regarding these biases potentially exploit LLM judge preferences. While our unmodified benchmark may favor models that provide more detailed responses, our style-controlled scores show no preference toward responses with enhanced styles or longer length over the original model's output (Table 5; Line 441). This suggests stylistic biases are effectively mitigated in our benchmark.
>
> **Self-Bias**
>
> We agree with the reviewer that LLM-as-a-Judge may also introduce self-biases (Line 488). In section 6.6, we proposed Ensemble LLM judges to mitigate self-biases. In our experiment, we observed that combining GPT-4-Turbo and Gemini-1.5-Pro judgments reduces self-biases exhibited in the benchmark using GPT-4-Turbo as a single judge (Line 497). Furthermore, our results showed that our ensemble method achieves higher agreement and correlation to human preferences than the single judge approach (Line 394). This suggests self-biases are mitigated in our benchmark.
>
> Lastly, we note that our benchmark achieves strong agreement and correlation with human preference, thereby validating our benchmark's quality and usefulness to the community (Table 1; Table 3).

---

> ### Author Response · Authors · 2024-11-18
> **Official Response by Authors**
>
> ### Reviewer's Question
>
> > Is there any estimation on the error/quality of the data generated? Or using some metrics to evaluate the similarity of the generated data with the real-world data?
>
> We are concerned there may be a potential misunderstanding. Our Bench-O-Matic pipeline does not generate data, but serves as a curation pipeline. It automatically identifies high-quality prompts from real-world data without modifying the prompts. Since we are applying the pipeline on real-world user prompts, the resulting datasets are still real-world user prompts. To evaluate the quality of the resulting dataset, we focus on how well it serves as a benchmark for predicting model performance with fidelity to human preferences. In our ablation, we found that unbiased random samples of real-world data produce low quality benchmarks while our targeted sample of the same real-world data is much higher in quality (Line 452).
>
> If the reviewer is referring to the preference labels generated by the LLM judge during model evaluation, we proposed metrics to measure how well a benchmark aligns with human preferences in Section 3. Our metrics demonstrate that the curated benchmarks strongly agree and correlate with human preference leaderboards (Table 1; Table 2; Table 3). These results implies that when evaluating a model, our benchmark can accurately predict its performance on real-world preference tasks, which is the primary goal of an effective benchmark.
>
> ---
>
> We thank the reviewer for their invaluable feedback, and we sincerely hope that the reviewer would reconsider their rating in light of all our points listed in the rebuttal.

---

> > ### Author Response · Authors · 2024-11-23
> > **Sincere Request for Review of Our Responses**
> >
> > As we approach the end of the discussion period, we want to ensure that our response has adequately addressed all of your concerns. Please advise if further clarification is needed or if there are additional questions. We are keen to address any remaining issues and hope you might reconsider your rating based on the information provided.
> >
> > Thank you for your time and consideration.
> >
> > Best Regards,
> >
> > Authors

---

> ### Author Response · Authors · 2024-12-02
> **Gentle Reminder**
>
> As we approach the end of the discussion period tonight, we want to ensure that our response has adequately addressed all of your concerns. Please advise if further clarification is needed or if there are additional questions. We are keen to address any remaining issues and hope you might reconsider your rating based on the information provided.
>
> Thank you for your time and consideration.
>
> Best Regards,
>
> Authors

---

### Meta-Review · Area_Chair_VwvK · 2024-12-21

**Metareview:**

The paper presents an automated pipeline, Bench-O-Matic, which curates prompts and creates benchmarks from large, crowd-sourced datasets for evaluating LLMs. Additionally, new metrics are proposed to assess benchmark quality. Experiments demonstrate that its performance aligns with human preferences. However, the reliability of the new benchmark relies heavily on LLMs’ judgments, and the current version does not address the generalizability of the pipeline. While I acknowledge that the pipeline can be applied to other data sources, generalizability is crucial if this work focuses on LLM evaluation.

**Additional Comments On Reviewer Discussion:**

Discussion Summary:

1. Clarification on LLMs’ Bias: The authors use an ensemble of judges to mitigate LLMs’ bias is partially effective. This approach is acceptable, as it is commonly adopted in similar works.
2. Generalization Ability of Bench-O-Matic: The authors claim that the pipeline can be applied to other data sources. However, throughout the rebuttal session, no new experiments were presented to support this claim.

---

### Decision · Program_Chairs · 2025-01-22

Reject